# Adversarially Robust Optimization
# with Gaussian Processes

**Ilija Bogunovic**
LIONS, EPFL
ilija.bogunovic@epfl.ch

**Jonathan Scarlett**
National University of Singapore
scarlett@comp.nus.edu.sg

**Stefanie Jegelka**
MIT CSAIL
stefje@mit.edu

**Volkan Cevher**
LIONS, EPFL
volkan.cevher@epfl.ch

## Abstract

In this paper, we consider the problem of Gaussian process (GP) optimization with an added robustness requirement: The returned point may be perturbed by an adversary, and we require the function value to remain as high as possible even after this perturbation. This problem is motivated by settings in which the underlying functions during optimization and implementation stages are different, or when one is interested in finding an entire region of good inputs rather than only a single point. We show that standard GP optimization algorithms do not exhibit the desired robustness properties, and provide a novel confidence-bound based algorithm STABLEOPT for this purpose. We rigorously establish the required number of samples for STABLEOPT to find a near-optimal point, and we complement this guarantee with an algorithm-independent lower bound. We experimentally demonstrate several potential applications of interest using real-world data sets, and we show that STABLEOPT consistently succeeds in finding a stable maximizer where several baseline methods fail.

## 1 Introduction

Gaussian processes (GP) provide a powerful means for sequentially optimizing a black-box function $f$ that is costly to evaluate and for which noisy point evaluations are available. Since its introduction, this approach has successfully been applied to numerous applications, including robotics [21], hyperparameter tuning [30], recommender systems [34], environmental monitoring [31], and more.

In many such applications, one is faced with various forms of uncertainty that are not accounted for by standard algorithms. In robotics, the optimization is often performed via simulations, creating a mismatch between the assumed function and the true one; in hyperparameter tuning, the function is typically similarly mismatched due to limited training data; in recommendation systems and several other applications, the underlying function is inherently time-varying, so the returned solution may become increasingly stale over time; the list goes on.

In this paper, we address these considerations by studying the GP optimization problem with an additional requirement of *adversarial robustness*: The returned point may be perturbed by an adversary, and we require the function value to remain as high as possible even after this perturbation. This problem is of interest not only for attaining improved robustness to uncertainty, but also for settings where one seeks a region of good points rather than a single point, and for other related max-min optimization settings (see Section 4 for further discussion).

**Related work.** Numerous algorithms have been developed for GP optimization in recent years [7, 16, 17, 26, 28, 31, 35]. Beyond the standard setting, several important extensions have been considered, including batch sampling [11, 12, 14], contextual and time-varying settings [6, 20], safety requirements [33], and high dimensional settings [18, 25, 36], just to name a few.

Various forms of robustness in GP optimization have been considered previously. A prominent example is that of outliers [22], in which certain function values are highly unreliable; however, this is a separate issue from that of the present paper, since in [22] the returned point does not undergo any perturbation. Another related recent work is [2], which assumes that the *sampled points* (rather than the returned one) are subject to uncertainty. In addition to this difference, the uncertainty in [2] is random rather than adversarial, which is complementary but distinct from our work. The same is true of a setting called *unscented Bayesian optimization* in [23]. Moreover, no theoretical results are given in [2, 23]. In [8], a robust form of batch optimization is considered, but with yet another form of robustness, namely, some experiments in the batch may fail to produce an outcome. Level-set estimation [7, 15] is another approach to finding regions of good points rather than a single point.

Our problem formulation is also related to other works on non-convex robust optimization, particularly those of *Bertsimas et al.* [3, 4]. In these works, a stable design $\mathbf{x}$ is sought that solves $\min_{\mathbf{x} \in D} \max_{\boldsymbol{\delta} \in \mathcal{U}} f(\mathbf{x} + \boldsymbol{\delta})$. Here, $\boldsymbol{\delta}$ resides in some uncertainty set $\mathcal{U}$, and represents the perturbation against which the design $\mathbf{x}$ needs to be protected. Related problems have also recently been considered in the context of adversarial training (e.g., [29]). Compared to these works, our work bears the crucial difference that the objective function is *unknown*, and we can only learn about it through noisy point evaluations (i.e. bandit feedback).

Other works, such as [5, 9, 19, 32, 37], have considered robust optimization problems of the following form: For a given set of objectives $\{f_1, \ldots, f_m\}$ find $\mathbf{x}$ achieving $\max_{\mathbf{x} \in D} \min_{i=1,\ldots,m} f_i(\mathbf{x})$. We discuss variations of our algorithm for this type of formulation in Section 4.

**Contributions.** We introduce a variant of GP optimization in which the returned solution is required to exhibit stability/robustness to an adversarial perturbation. We demonstrate the failures of standard algorithms, and introduce a new algorithm STABLEOPT that overcomes these limitations. We provide a novel theoretical analysis characterizing the number of samples required for STABLEOPT to attain a near-optimal robust solution, and we complement this with an algorithm-independent lower bound. We provide several variations of our max-min optimization framework and theory, including connections and comparisons to previous works. Finally, we experimentally demonstrate a variety of potential applications of interest using real-world data sets, and we show that STABLEOPT consistently succeeds in finding a stable maximizer where several baseline methods fail.

## 2 Problem Setup

**Model.** Let $f$ be an unknown reward function over a domain $D \subseteq \mathbb{R}^p$ for some dimension $p$. At time $t$, we query $f$ at a single point $\mathbf{x}_t \in D$ and observe a noisy sample $y_t = f(\mathbf{x}_t) + z_t$, where $z_t \sim \mathcal{N}(0, \sigma^2)$. After $T$ rounds, a recommended point $\mathbf{x}^{(T)}$ is returned. In contrast with the standard goal of making $f(\mathbf{x}^{(T)})$ as high as possible, we seek to find a point such that $f$ remains high even after an adversarial perturbation; a formal description is given below.

We assume that $D$ is endowed with a kernel function $k(\cdot, \cdot)$, and $f$ has a bounded norm in the corresponding Reproducing Kernel Hilbert Space (RKHS) $\mathcal{H}_k(D)$. Specifically, we assume that $f \in \mathcal{F}_k(B)$, where

$$\mathcal{F}_k(B) = \{f \in \mathcal{H}_k(D) : \|f\|_k \leq B\}, \tag{1}$$

and $\|f\|_k$ is the RKHS norm in $\mathcal{H}_k(D)$. It is well-known that this assumption permits the construction of confidence bounds via Gaussian process (GP) methods; see Lemma 1 below for a precise statement. We assume that the kernel is normalized to satisfy $k(\mathbf{x}, \mathbf{x}) = 1$ for all $\mathbf{x} \in D$. Two commonly-considered kernels are squared exponential (SE) and Matérn:

$$k_{\text{SE}}(\mathbf{x}, \mathbf{x}') = \exp\left(-\frac{\|\mathbf{x} - \mathbf{x}'\|^2}{2l^2}\right), \tag{2}$$

$$k_{\text{Mat}}(\mathbf{x}, \mathbf{x}') = \frac{2^{1-\nu}}{\Gamma(\nu)}\left(\frac{\sqrt{2\nu}\|\mathbf{x} - \mathbf{x}'\|}{l}\right)J_\nu\left(\frac{\sqrt{2\nu}\|\mathbf{x} - \mathbf{x}'\|}{l}\right), \tag{3}$$

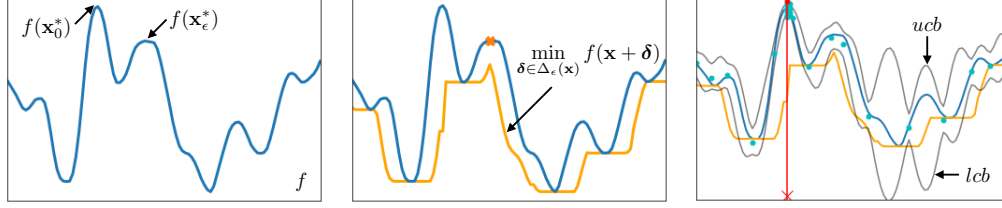

Figure 1: (Left) A function $f$ and its maximizer $\mathbf{x}_0^*$. (Middle) For $\epsilon = 0.06$ and $d(x, x') = |x - x'|$, the decision that corresponds to the local "wider" maximum of $f$ is the *optimal $\epsilon$-stable* decision. (Right) GP-UCB selects a point that nearly maximizes $f$, but is suboptimal in the $\epsilon$-stable sense.

where $l$ denotes the length-scale, $\nu > 0$ is an additional parameter that dictates the smoothness, and $J(\nu)$ and $\Gamma(\nu)$ denote the modified Bessel function and the gamma function, respectively [24].

Given a sequence of decisions $\{\mathbf{x}_1, \cdots, \mathbf{x}_t\}$ and their noisy observations $\{y_1, \cdots, y_t\}$, the posterior distribution under a $\mathrm{GP}(0, k(\mathbf{x}, \mathbf{x}'))$ prior is also Gaussian, with the following mean and variance:

$$\mu_t(\mathbf{x}) = \mathbf{k}_t(\mathbf{x})^T \big(\mathbf{K}_t + \sigma^2 \mathbf{I}\big)^{-1} \mathbf{y}_t, \tag{4}$$

$$\sigma_t^2(\mathbf{x}) = k(\mathbf{x}, \mathbf{x}) - \mathbf{k}_t(\mathbf{x})^T \big(\mathbf{K}_t + \sigma^2 \mathbf{I}\big)^{-1} \mathbf{k}_t(\mathbf{x}), \tag{5}$$

where $\mathbf{k}_t(\mathbf{x}) = \big[k(\mathbf{x}_i, \mathbf{x})\big]_{i=1}^{t}$, and $\mathbf{K}_t = \big[k(\mathbf{x}_t, \mathbf{x}_{t'})\big]_{t, t'}$ is the kernel matrix.

**Optimization goal.** Let $d(\mathbf{x}, \mathbf{x}')$ be a function mapping $D \times D \to \mathbb{R}$, and let $\epsilon$ be a constant known as the *stability parameter*. For each point $\mathbf{x} \in D$, we define a set

$$\Delta_\epsilon(\mathbf{x}) = \big\{\mathbf{x}' - \mathbf{x} : \mathbf{x}' \in D \text{ and } d(\mathbf{x}, \mathbf{x}') \leq \epsilon\big\}. \tag{6}$$

One can interpret this as the set of perturbations of $\mathbf{x}$ such that the newly obtained point $\mathbf{x}'$ is within a "distance" $\epsilon$ of $\mathbf{x}$. While we refer to $d(\cdot, \cdot)$ as the distance function throughout the paper, we allow it to be a general function, and not necessarily a distance in the mathematical sense. As we exemplify in Section 5, the parameter $\epsilon$ might be naturally specified as part of the application, or might be better treated as a parameter that can be tuned for the purpose of the overall learning goal.

We define an $\epsilon$-*stable optimal input* to be any $\mathbf{x}_\epsilon^*$ satisfying

$$\mathbf{x}_\epsilon^* \in \arg\max_{\mathbf{x} \in D} \min_{\boldsymbol{\delta} \in \Delta_\epsilon(\mathbf{x})} f(\mathbf{x} + \boldsymbol{\delta}). \tag{7}$$

Our goal is to report a point $\mathbf{x}^{(T)}$ that is stable in the sense of having low $\epsilon$-*regret*, defined as

$$r_\epsilon(\mathbf{x}) = \min_{\boldsymbol{\delta} \in \Delta_\epsilon(\mathbf{x}_\epsilon^*)} f(\mathbf{x}_\epsilon^* + \boldsymbol{\delta}) - \min_{\boldsymbol{\delta} \in \Delta_\epsilon(\mathbf{x})} f(\mathbf{x} + \boldsymbol{\delta}). \tag{8}$$

Note that once $r_\epsilon(\mathbf{x}) \leq \eta$ for some accuracy value $\eta \geq 0$, it follows that

$$\min_{\boldsymbol{\delta} \in \Delta_\epsilon(\mathbf{x})} f(\mathbf{x} + \boldsymbol{\delta}) \geq \min_{\boldsymbol{\delta} \in \Delta_\epsilon(\mathbf{x}_\epsilon^*)} f(\mathbf{x}_\epsilon^* + \boldsymbol{\delta}) - \eta. \tag{9}$$

We assume that $d(\cdot, \cdot)$ and $\epsilon$ are known, i.e., they are specified as part of the optimization formulation.

As a running example, we consider the case that $d(\mathbf{x}, \mathbf{x}') = \|\mathbf{x} - \mathbf{x}'\|$ for some norm $\|\cdot\|$ (e.g., $\ell_2$-norm), in which case achieving low $\epsilon$-regret amounts to favoring *broad peaks* instead of narrow ones, particularly for higher $\epsilon$; see Figure 1 for an illustration. In Section 4, we discuss how our framework also captures a variety of other max-min optimization settings of interest.

**Failure of classical methods.** Various algorithms have been developed for achieving small regret in the standard GP optimization problem. A prominent example is GP-UCB, which chooses

$$\mathbf{x}_t \in \arg\max_{\mathbf{x} \in D} \mathrm{ucb}_{t-1}(\mathbf{x}), \tag{10}$$

where $\mathrm{ucb}_{t-1}(\mathbf{x}) := \mu_{t-1}(\mathbf{x}) + \beta_t^{1/2} \sigma_{t-1}(\mathbf{x})$. This algorithm is guaranteed to achieve sublinear cumulative regret with high probability [31], for a suitably chosen $\beta_t$. While this is useful when

**Algorithm 1** The STABLEOPT algorithm

---

**Input:** Domain $D$, GP prior $(\mu_0, \sigma_0, k)$, parameters $\{\beta_t\}_{t \geq 1}$, stability $\epsilon$, distance function $d(\cdot, \cdot)$
1: **for** $t = 1, 2, \ldots, T$ **do**
2:     Set

$$\tilde{\mathbf{x}}_t = \arg\max_{\mathbf{x} \in D} \min_{\boldsymbol{\delta} \in \Delta_\epsilon(\mathbf{x})} \text{ucb}_{t-1}(\mathbf{x} + \boldsymbol{\delta}). \tag{13}$$

3:     Set $\boldsymbol{\delta}_t = \arg\min_{\boldsymbol{\delta} \in \Delta_\epsilon(\tilde{\mathbf{x}}_t)} \text{lcb}_{t-1}(\tilde{\mathbf{x}}_t + \boldsymbol{\delta})$
4:     Sample $\tilde{\mathbf{x}}_t + \boldsymbol{\delta}_t$, and observe $y_t = f(\tilde{\mathbf{x}}_t + \boldsymbol{\delta}_t) + z_t$
5:     Update $\mu_t, \sigma_t, \text{ucb}_t$ and $\text{lcb}_t$ according to (5) and (12), by including $\{(\tilde{\mathbf{x}}_t + \boldsymbol{\delta}_t, y_t)\}$
6: **end for**

---

$\mathbf{x}_\epsilon^* = \mathbf{x}_0^*$;[1] in general for a given fixed $\epsilon \neq 0$, these two decisions may not coincide, and hence, $\min_{\boldsymbol{\delta} \in \Delta_\epsilon(\mathbf{x}_0^*)} f(\mathbf{x}_0^* + \boldsymbol{\delta})$ can be significantly smaller than $\min_{\boldsymbol{\delta} \in \Delta_\epsilon(\mathbf{x}_\epsilon^*)} f(\mathbf{x}_\epsilon^* + \boldsymbol{\delta})$.

A visual example is given in Figure 1 (Right), where the selected point of GP-UCB for $t = 20$ is shown. This point nearly maximizes $f$, but it is strictly suboptimal in the $\epsilon$-stable sense. The same limitation applies to other GP optimization strategies (e.g., [7, 16, 17, 26, 28, 35]) whose goal is to identify the global non-robust maximum $\mathbf{x}_0^*$. In Section 5, we will see that more advanced baseline strategies also perform poorly when applied to our problem.

## 3 Proposed Algorithm and Theory

Our proposed algorithm, STABLEOPT, is described in Algorithm 1, and makes use of the following confidence bounds depending on an *exploration parameter* $\beta_t$ (*cf.*, Lemma 1 below):

$$\text{ucb}_{t-1}(\mathbf{x}) := \mu_{t-1}(\mathbf{x}) + \beta_t^{1/2} \sigma_{t-1}(\mathbf{x}), \tag{11}$$

$$\text{lcb}_{t-1}(\mathbf{x}) := \mu_{t-1}(\mathbf{x}) - \beta_t^{1/2} \sigma_{t-1}(\mathbf{x}). \tag{12}$$

The point $\tilde{\mathbf{x}}_t$ defined in (13) is the one having the highest "stable" upper confidence bound. However, the queried point is not $\tilde{\mathbf{x}}_t$, but instead $\tilde{\mathbf{x}}_t + \boldsymbol{\delta}_t$, where $\boldsymbol{\delta}_t \in \Delta_\epsilon(\tilde{\mathbf{x}}_t)$ is chosen to minimize the *lower* confidence bound. As a result, the algorithm is based on two distinct principles: (i) optimism in the face of uncertainty when it comes to selecting $\tilde{\mathbf{x}}_t$; (ii) pessimism in the face of uncertainty when it comes to anticipating the perturbation of $\tilde{\mathbf{x}}_t$. The first of these is inherent to existing algorithms such as GP-UCB [31], whereas the second is unique to the adversarially robust GP optimization problem. An example illustration of STABLEOPT's execution is given in the supplementary material.

We have left the final reported point $\mathbf{x}^{(T)}$ unspecified in Algorithm 1, as there are numerous reasonable choices. The simplest choice is to simply return $\mathbf{x}^{(T)} = \tilde{\mathbf{x}}_T$, but in our theory and experiments, we will focus on $\mathbf{x}^{(T)}$ equaling the point in $\{\tilde{\mathbf{x}}_1, \ldots, \tilde{\mathbf{x}}_T\}$ with the highest lower confidence bound.

Finding an exact solution to the optimization of the acquisition function in (13) can be challenging in practice. When $D$ is continuous, a natural approach is to find an approximate solution using an efficient local search algorithm for robust optimization with a fully known objective function, such as that of [4].

### 3.1 Upper bound on $\epsilon$-regret

Our analysis makes use of the *maximum information gain* under $t$ noisy measurements:

$$\gamma_t = \max_{\mathbf{x}_1, \cdots, \mathbf{x}_t} \frac{1}{2} \log \det(\mathbf{I}_t + \sigma^{-2} \mathbf{K}_t), \tag{14}$$

which has been used in numerous theoretical works on GP optimization following [31].

STABLEOPT depends on the exploration parameter $\beta_t$, which determines the width of the confidence bounds. In our main result, we set $\beta_t$ as in [10] and make use of the following.

**Lemma 1.** [10] *Fix $f \in \mathcal{F}_k(B)$, and consider the sampling model $y_t = f(\mathbf{x}_t) + z_t$ with $z_t \sim \mathcal{N}(0, \sigma^2)$, with independence between times. Under the choice $\beta_t = \left(B + \sigma\sqrt{2(\gamma_{t-1} + \log\frac{e}{\xi})}\right)^2$, the following holds with probability at least $1 - \xi$:*

$$\text{lcb}_{t-1}(\mathbf{x}) \leq f(\mathbf{x}) \leq \text{ucb}_{t-1}(\mathbf{x}), \quad \forall \mathbf{x} \in D, \forall t \geq 1. \qquad (15)$$

The following theorem bounds the performance of STABLEOPT under a suitable choice of the recommended point $\mathbf{x}^{(T)}$. The proof is given in the supplementary material.

**Theorem 1.** (Upper Bound) *Fix $\epsilon > 0$, $\eta > 0$, $B > 0$, $T \in \mathbb{Z}$, $\xi \in (0, 1)$, and a distance function $d(\mathbf{x}, \mathbf{x}')$, and suppose that*

$$\frac{T}{\beta_T \gamma_T} \geq \frac{C_1}{\eta^2}, \qquad (16)$$

*where $C_1 = 8/\log(1 + \sigma^{-2})$. For any $f \in \mathcal{F}_k(B)$, STABLEOPT with $\beta_t$ set as in Lemma 1 achieves $r_\epsilon(\mathbf{x}^{(T)}) \leq \eta$ after $T$ rounds with probability at least $1 - \xi$, where*

$$\mathbf{x}^{(T)} = \tilde{\mathbf{x}}_{t^*}, \qquad t^* = \underset{t=1,\ldots,T}{\arg\max} \min_{\boldsymbol{\delta} \in \Delta_\epsilon(\tilde{\mathbf{x}}_t)} \text{lcb}_{t-1}(\tilde{\mathbf{x}}_t + \boldsymbol{\delta}). \qquad (17)$$

This result holds for general kernels, and for both finite and continuous $D$. Our analysis bounds function values according to the confidence bounds in Lemma 1 analogously to GP-UCB [31], but also addresses the non-trivial challenge of characterizing the perturbations $\boldsymbol{\delta}_t$. While we focused on the non-Bayesian RKHS setting, the proof can easily be adapted to handle the *Bayesian optimization* (BO) setting in which $f \sim \text{GP}(0, k)$; see Section 4 for further discussion.

Theorem 1 can be made more explicit by substituting bounds on $\gamma_T$; in particular, $\gamma_T = O((\log T)^{p+1})$ for the SE kernel, and $\gamma_T = O(T^{\frac{p(p+1)}{2\nu + p(p+1)}} \log T)$ for the Matérn-$\nu$ kernel [31]. The former yields $T = O^*\left(\frac{1}{\eta^2}\left(\log\frac{1}{\eta}\right)^{2p}\right)$ in Theorem 1 for constant $B$, $\sigma^2$, and $\epsilon$ (where $O^*(\cdot)$ hides dimension-independent log factors), which we will shortly see nearly matches an algorithm-independent lower bound.

## 3.2 Lower bound on $\epsilon$-regret

Establishing lower bounds under general kernels and input domains is an open problem even in the non-robust setting. Accordingly, the following theorem focuses on a more specific setting than the upper bound: We let the input domain be $[0, 1]^p$ for some dimension $p$, and we focus on the SE and Matérn kernels. In addition, we only consider the case that $d(\mathbf{x}, \mathbf{x}') = \|\mathbf{x} - \mathbf{x}'\|_2$, though extensions to other norms (e.g., $\ell_1$ or $\ell_\infty$) follow immediately from the proof.

**Theorem 2.** (Lower Bound) *Let $D = [0, 1]^p$ for some dimension $p$, and set $d(\mathbf{x}, \mathbf{x}') = \|\mathbf{x} - \mathbf{x}'\|_2$. Fix $\epsilon \in \left(0, \frac{1}{2}\right)$, $\eta \in \left(0, \frac{1}{2}\right)$, $B > 0$, and $T \in \mathbb{Z}$. Suppose there exists an algorithm that, for any $f \in \mathcal{F}_k(B)$, reports a point $\mathbf{x}^{(T)}$ achieving $\epsilon$-regret $r_\epsilon(\mathbf{x}^{(T)}) \leq \eta$ after $T$ rounds with probability at least $1 - \xi$. Then, provided that $\frac{\eta}{B}$ and $\xi$ are sufficiently small, we have the following:*

1. *For $k = k_{\text{SE}}$, it is necessary that $T = \Omega\left(\frac{\sigma^2}{\eta^2}\left(\log\frac{B}{\eta}\right)^{p/2}\right)$.*

2. *For $k = k_{\text{Matérn}}$, it is necessary that $T = \Omega\left(\frac{\sigma^2}{\eta^2}\left(\frac{B}{\eta}\right)^{p/\nu}\right)$.*

*Here we assume that the stability parameter $\epsilon$, dimension $p$, target probability $\xi$, and kernel parameters $l, \nu$ are fixed (i.e., not varying as a function of the parameters $T$, $\eta$, $\sigma$ and $B$).*

The proof is based on constructing a finite subset of "difficult" functions in $\mathcal{F}_k(B)$ and applying lower bounding techniques from the multi-armed bandit literature, also making use of several auxiliary results from the non-robust setting [27]. More specifically, the functions in the restricted class consist of narrow negative "valleys" that the adversary can perturb the reported point into, but that are hard to identify until a large number of samples have been taken.

For constant $\sigma^2$ and $B$, the condition for the SE kernel simplifies to $T = \Omega\left(\frac{1}{\eta^2}\left(\log\frac{1}{\eta}\right)^{p/2}\right)$, thus nearly matching the upper bound $T = O^*\left(\frac{1}{\eta^2}\left(\log\frac{1}{\eta}\right)^{2p}\right)$ of STABLEOPT established above. In the case of the Matérn kernel, more significant gaps remain between the upper and lower bounds; however, similar gaps remain even in the standard (non-robust) setting [27].

# 4 Variations of STABLEOPT

While the above problem formulation seeks robustness within an $\epsilon$-ball corresponding to the distance function $d(\cdot, \cdot)$, our algorithm and theory apply to a variety of seemingly distinct settings. We outline a few such settings here; in the supplementary material, we give details of their derivations.

**Robust Bayesian optimization.** Algorithm 1 and Theorem 1 extend readily to the Bayesian setting in which $f \sim \mathrm{GP}(0, k(\mathbf{x}, \mathbf{x}'))$. In particular, since the proof of Theorem 1 is based on confidence bounds, the only change required is selecting $\beta_t$ to be that used for the Bayesian setting in [31]. As a result, our framework also captures the novel problem of *adversarially robust Bayesian optimization*. All of the variations outlined below similarly apply to both the Bayesian and non-Bayesian settings.

**Robustness to unknown parameters.** Consider the scenario where an unknown function $f : D \times \Theta \to \mathbb{R}$ has a bounded RKHS norm under some composite kernel $k((\mathbf{x}, \boldsymbol{\theta}), (\mathbf{x}', \boldsymbol{\theta}'))$. Some special cases include $k((\mathbf{x}, \boldsymbol{\theta}), (\mathbf{x}', \boldsymbol{\theta}')) = k(\mathbf{x}, \mathbf{x}') + k(\boldsymbol{\theta}, \boldsymbol{\theta}')$ and $k((\mathbf{x}, \boldsymbol{\theta}), (\mathbf{x}', \boldsymbol{\theta}')) = k(\mathbf{x}, \mathbf{x}')k(\boldsymbol{\theta}, \boldsymbol{\theta}')$ [20]. The posterior mean $\mu_t(\mathbf{x}, \boldsymbol{\theta})$ and variance $\sigma_t^2(\mathbf{x}, \boldsymbol{\theta})$ conditioned on the previous observations $(\mathbf{x}_1, \boldsymbol{\theta}_1, y_1), ..., (\mathbf{x}_{t-1}, \boldsymbol{\theta}_{t-1}, y_{t-1})$ are computed analogously to (5) [20].

A robust optimization formulation in this setting is to seek $\mathbf{x}$ that solves

$$\max_{\mathbf{x} \in D} \min_{\boldsymbol{\theta} \in \Theta} f(\mathbf{x}, \boldsymbol{\theta}). \tag{18}$$

That is, we seek to find a configuration $\mathbf{x}$ that is robust against any possible parameter vector $\boldsymbol{\theta} \in \Theta$.

Potential applications of this setup include the following:

- In areas such a robotics, we may have numerous parameters to tune (given by $\mathbf{x}$ and $\boldsymbol{\theta}$ collectively), but when it comes to implementation, some of them (i.e., only $\boldsymbol{\theta}$) become out of our control. Hence, we need to be robust against whatever values they may take.
- We wish to tune hyperparameters in order to make an algorithm work simultaneously for a number of distinct data types that bear some similarities/correlations. The data types are represented by $\boldsymbol{\theta}$, and we can choose the data type to our liking during the optimization stage.

STABLEOPT can be used to solve (18); we maintain $\boldsymbol{\theta}_t$ instead of $\boldsymbol{\delta}_t$, and modify the main steps to

$$\mathbf{x}_t \in \arg\max_{\mathbf{x} \in D} \min_{\boldsymbol{\theta} \in \Theta} \mathrm{ucb}_{t-1}(\mathbf{x}, \boldsymbol{\theta}), \tag{19}$$

$$\boldsymbol{\theta}_t \in \arg\min_{\boldsymbol{\theta} \in \Theta} \mathrm{lcb}_{t-1}(\mathbf{x}_t, \boldsymbol{\theta}). \tag{20}$$

At each time step, STABLEOPT receives a noisy observation $y_t = f(\mathbf{x}_t, \boldsymbol{\theta}_t) + z_t$, which is used with $(\mathbf{x}_t, \boldsymbol{\theta}_t)$ for computing the posterior. As explained in the supplementary material, the guarantee $r_\epsilon(\mathbf{x}^{(T)}) \leq \eta$ in Theorem 1 is replaced by $\min_{\boldsymbol{\theta} \in \Theta} f(\mathbf{x}^{(T)}, \boldsymbol{\theta}) \geq \max_{\mathbf{x} \in D} \min_{\boldsymbol{\theta} \in \Theta} f(\mathbf{x}, \boldsymbol{\theta}) - \eta$.

**Robust estimation.** Continuing with the consideration of a composite kernel on $(\mathbf{x}, \boldsymbol{\theta})$, we consider the following practical problem variant proposed in [4]. Let $\bar{\boldsymbol{\theta}} \in \Theta$ be an estimate of the true problem coefficient $\boldsymbol{\theta}^* \in \Theta$. Since, $\bar{\boldsymbol{\theta}}$ is an estimate, the true coefficient satisfies $\boldsymbol{\theta}^* = \bar{\boldsymbol{\theta}} + \boldsymbol{\delta}_{\boldsymbol{\theta}}$, where $\boldsymbol{\delta}_{\boldsymbol{\theta}}$ represents uncertainty in $\bar{\boldsymbol{\theta}}$. Often, practitioners solve $\max_{\mathbf{x} \in D} f(\mathbf{x}, \bar{\boldsymbol{\theta}})$ and ignore the uncertainty. As a more sophisticated approach, we let $\Delta_\epsilon(\bar{\boldsymbol{\theta}}) = \{\boldsymbol{\theta} - \bar{\boldsymbol{\theta}} : \boldsymbol{\theta} \in \Theta \text{ and } d(\bar{\boldsymbol{\theta}}, \boldsymbol{\theta}) \leq \epsilon\}$, and consider the following robust problem formulation:

$$\max_{\mathbf{x} \in D} \min_{\boldsymbol{\delta}_{\boldsymbol{\theta}} \in \Delta_\epsilon(\bar{\boldsymbol{\theta}})} f(\mathbf{x}, \bar{\boldsymbol{\theta}} + \boldsymbol{\delta}_{\boldsymbol{\theta}}). \tag{21}$$

For the given estimate $\bar{\boldsymbol{\theta}}$, the main steps of STABLEOPT in this setting are

$$\mathbf{x}_t \in \arg\max_{\mathbf{x} \in D} \min_{\boldsymbol{\delta}_{\boldsymbol{\theta}} \in \Delta_\epsilon(\bar{\boldsymbol{\theta}})} \mathrm{ucb}_{t-1}(\mathbf{x}, \bar{\boldsymbol{\theta}} + \boldsymbol{\delta}_{\boldsymbol{\theta}}), \tag{22}$$

$$\boldsymbol{\delta}_{\boldsymbol{\theta}, t} \in \arg\min_{\boldsymbol{\delta}_{\boldsymbol{\theta}} \in \Delta_\epsilon(\bar{\boldsymbol{\theta}})} \mathrm{lcb}_{t-1}(\mathbf{x}_t, \bar{\boldsymbol{\theta}} + \boldsymbol{\delta}_{\boldsymbol{\theta}}), \tag{23}$$

and the noisy observations take the form $y_t = f(\mathbf{x}_t, \bar{\boldsymbol{\theta}} + \boldsymbol{\delta}_{\boldsymbol{\theta}, t}) + z_t$. The guarantee $r_\epsilon(\mathbf{x}^{(T)}) \leq \eta$ in Theorem 1 is replaced by $\min_{\boldsymbol{\delta}_{\boldsymbol{\theta}} \in \Delta_\epsilon(\bar{\boldsymbol{\theta}})} f(\mathbf{x}^{(T)}, \bar{\boldsymbol{\theta}} + \boldsymbol{\delta}_{\boldsymbol{\theta}}) \geq \max_{\mathbf{x} \in D} \min_{\boldsymbol{\delta}_{\boldsymbol{\theta}} \in \Delta_\epsilon(\bar{\boldsymbol{\theta}})} f(\mathbf{x}, \bar{\boldsymbol{\theta}} + \boldsymbol{\delta}_{\boldsymbol{\theta}}) - \eta$.

**Robust group identification.** In some applications, it is natural to partition $D$ into disjoint subsets, and search for the subset with the highest worst-case function value (see Section 5 for a movie

recommendation example). Letting $\mathcal{G} = \{G_1, \dots, G_k\}$ denote the groups that partition the input space, the robust optimization problem is given by

$$\max_{G \in \mathcal{G}} \min_{\mathbf{x} \in G} f(\mathbf{x}), \qquad (24)$$

and the algorithm reports a group $G^{(T)}$. The main steps of STABLEOPT are given by

$$G_t \in \arg\max_{G \in \mathcal{G}} \min_{\mathbf{x} \in G} \mathrm{ucb}_{t-1}(\mathbf{x}), \qquad (25)$$

$$\mathbf{x}_t \in \arg\min_{\mathbf{x} \in G_t} \mathrm{lcb}_{t-1}(\mathbf{x}), \qquad (26)$$

and the observations are of the form $y_t = f(\mathbf{x}_t) + z_t$ as usual. The guarantee $r_\epsilon(\mathbf{x}^{(T)}) \leq \eta$ in Theorem 1 is replaced by $\min_{\mathbf{x} \in G^{(T)}} f(\mathbf{x}) \geq \max_{G \in \mathcal{G}} \min_{\mathbf{x} \in G} f(\mathbf{x}) - \eta$.

The preceding variations of STABLEOPT, as well as their resulting variations of Theorem 1, follow by substituting certain (unconventional) choices of $d(\cdot, \cdot)$ and $\epsilon$ into Algorithm 1 and Theorem 1, with $(\mathbf{x}, \boldsymbol{\theta})$ in place of $\mathbf{x}$ where appropriate. The details are given in the supplementary material.

## 5   Experiments

In this section, we experimentally validate the performance of STABLEOPT by comparing against several baselines. Each algorithm that we consider may distinguish between the *sampled point* (i.e., the point that produces the noisy observation $y_t$) and the *reported point* (i.e., the point believed to be near-optimal in terms of $\epsilon$-stability). For STABLEOPT, as described in Algorithm 1, the sampled point is $\tilde{\mathbf{x}}_t + \boldsymbol{\delta}_t$, and the reported point after time $t$ is the one in $\{\tilde{\mathbf{x}}_1, \dots, \tilde{\mathbf{x}}_t\}$ with the highest value of $\min_{\boldsymbol{\delta} \in \Delta_\epsilon(\tilde{\mathbf{x}}_t)} \mathrm{lcb}_t(\tilde{\mathbf{x}}_t + \boldsymbol{\delta})$.[2] In addition, we consider the following baselines:

- GP-UCB (see (10)). We consider GP-UCB to be a good representative of the wide range of existing methods that search for the non-robust global maximum.

- MAXIMIN-GP-UCB. We consider a natural generalization of GP-UCB in which, at each time step, the sampled and reported point are both given by

$$\mathbf{x}_t = \arg\max_{\mathbf{x} \in D} \min_{\boldsymbol{\delta} \in \Delta_\epsilon(x)} \mathrm{ucb}_{t-1}(\mathbf{x} + \boldsymbol{\delta}). \qquad (27)$$

- STABLE-GP-RANDOM. The sampling point $\mathbf{x}_t$ at every time step is chosen uniformly at random, while the reported point at time $t$ is chosen to be the point among the sampled points $\{\mathbf{x}_1, \dots, \mathbf{x}_t\}$ according to the same rule as the one used for STABLEOPT.

- STABLE-GP-UCB. The sampled point is given by the GP-UCB rule, while the reported point is again chosen in the same way as in STABLEOPT.

As observed in existing works (e.g., [7,31]), the theoretical choice of $\beta_t$ is overly conservative. We therefore adopt a constant value of $\beta_t^{1/2} = 2.0$ in each of the above methods, which we found to provide a suitable exploration/exploitation trade-off for each of the above algorithms.

Given a reported point $\mathbf{x}^{(t)}$ at time $t$, the performance metric is the $\epsilon$-regret $r_\epsilon(\mathbf{x}^{(t)})$ given in (8). Two observations are in order: (i) All the baselines are heuristic approaches for our problem, and they do not have guarantees in terms of near-optimal stability; (ii) We do not compare against other standard GP optimization methods, as their performance is comparable to that of GP-UCB; in particular, they suffer from exactly the same pitfalls described at the end of Section 2.

**Synthetic function.** We consider the synthetic function from [4] (see Figure 2a), given by

$$\begin{aligned} f_{\mathrm{poly}}(x, y) = &-2x^6 + 12.2x^5 - 21.2x^4 - 6.2x + 6.4x^3 + 4.7x^2 - y^6 + 11y^5 \\ &- 43.3y^4 + 10y + 74.8y^3 - 56.9y^2 + 4.1xy + 0.1y^2x^2 - 0.4y^2x - 0.4x^2y. \end{aligned} \qquad (28)$$

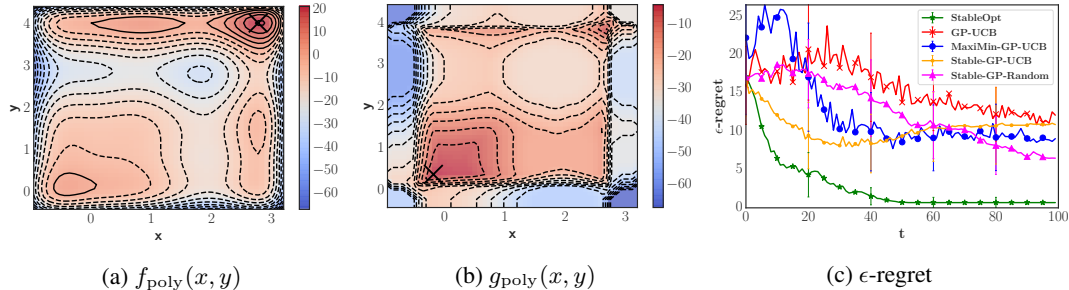

(a) $f_{\text{poly}}(x, y)$      (b) $g_{\text{poly}}(x, y)$      (c) $\epsilon$-regret

Figure 2: (Left) Synthetic function from [4]. (Middle) Counterpart with worst-case perturbations. (Right) The performance. In this example, STABLEOPT significantly outperforms the baselines.

The decision space is a uniformly spaced grid of points in $((-0.95, 3.2), (-0.45, 4.4))$ of size $10^4$. There exist multiple local maxima, and the global maximum is at $(x_f^*, y_f^*) = (2.82, 4.0)$, with $f_{\text{poly}}(x_f^*, y_f^*) = 20.82$. Similarly as in [4], given stability parameters $\boldsymbol{\delta} = (\delta_x, \delta_y)$, where $\|\boldsymbol{\delta}\|_2 \leq 0.5$, the robust optimization problem is

$$\max_{(x,y) \in D} g_{\text{poly}}(x, y), \tag{29}$$

where

$$g_{\text{poly}}(x, y) := \min_{(\delta_x, \delta_y) \in \Delta_{0.5}(x,y)} f_{\text{poly}}(x - \delta_x, y - \delta_y). \tag{30}$$

A plot of $g_{\text{poly}}$ is shown in Figure 2b. The global maximum is attained at $(x_g^*, y_g^*) = (-0.195, 0.284)$ and $g_{\text{poly}}(x_g^*, y_g^*) = -4.33$, and the inputs maximizing $f$ yield $g_{\text{poly}}(x_f^*, y_f^*) = -22.34$.

We initialize the above algorithms by selecting 10 uniformly random inputs $(x, y)$, keeping those points the same for each algorithm. The kernel adopted is a squared exponential ARD kernel. We randomly sample 500 points whose function value is above $-15.0$ to learn the GP hyperparameters via maximum likelihood, and then run the algorithms with these hyperparameters. The observation noise standard deviation is set to $0.1$, and the number of sampling rounds is $T = 100$. We repeat the experiment 100 times and show the average performance in Figure 2c. We observe that STA-BLEOPT significantly outperforms the baselines in this experiment. In the later rounds, the baselines report points that are close to the global optimizer, which is suboptimal with respect to the $\epsilon$-regret.

**Lake data.** In the supplementary material, we provide an analogous experiment to that above using chlorophyll concentration data from Lake Zürich, with STABLEOPT again performing best.

**Robust robot pushing.** We consider the deterministic version of the robot pushing objective from [35], with publicly available code.[3] The goal is to find a good pre-image for pushing an object to a target location. The 3-dimensional function takes as input the robot location $(r_x, r_y)$ and pushing duration $r_t$, and outputs $f(r_x, r_y, r_t) = 5 - d_{\text{end}}$, where $d_{\text{end}}$ is the distance from the pushed object to the target location. The domain $D$ is continuous: $r_x, r_y \in [-5, 5]$ and $r_t \in [1, 30]$.

We consider a twist on this problem in which there is uncertainty regarding the precise target location, so one seeks a set of input parameters that is robust against a number of different potential locations. In the simplest case, the number of such locations is finite, meaning we can model this problem as $\boldsymbol{r} \in \arg\max_{\boldsymbol{r} \in D} \min_{i \in [m]} f_i(\boldsymbol{r})$, where each $f_i$ corresponds to a different target location, and $[m] = \{1, \ldots, m\}$. This is a special case of (18) with a finite set $\Theta$ of cardinality $m$.

In our experiment, we use $m = 2$. Hence, our goal is to find an input configuration $\boldsymbol{r}$ that is robust against two different target locations. The first target is uniform over the domain, and the second is uniform over the $\ell_1$-ball centered at the first target location with radius $r = 2.0$. We initialize each algorithm with one random sample from each $f_i$. We run each method for $T = 100$ rounds, and for a reported point $\boldsymbol{r}_t$ at time $t$, we compare the methods in terms of the robust objective $\min_{i \in [m]} f_i(\boldsymbol{r}_t)$. We perform a fully Bayesian treatment of the hyperparameters, sampling every 10 rounds as in [17, 35]. We average over 30 random pairs of $\{f_1, f_2\}$ and report the results in Figure 3. STABLEOPT noticeably outperforms its competitors except in some of the very early rounds. We note that the apparent discontinuities in certain curves are a result of the hyperparmeter re-estimation.

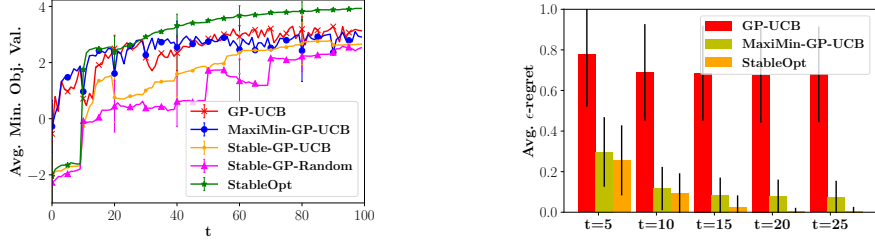

Figure 3: Robust robot pushing experiment (Left) and MovieLens-100K experiment (Right)

**Group movie recommendation.** Our goal in this task is to recommend a group of movies to a user such that *every* movie in the group is to their liking. We use the MovieLens-100K dataset, which consists of 1682 movies and 943 users. The data takes the form of an incomplete matrix $\mathbf{R}$ of ratings, where $R_{i,j}$ is the rating of movie $i$ given by the user $j$. To impute the missing rating values, we apply non-negative matrix factorization with $p = 15$ latent factors. This produces a feature vector for each movie $\mathbf{m}_i \in \mathbb{R}^p$ and user $\mathbf{u}_j \in \mathbb{R}^p$. We use $10\%$ of the user data for training, in which we fit a Gaussian distribution $P(\mathbf{u}) = \mathcal{N}(\mathbf{u}|\boldsymbol{\mu}, \boldsymbol{\Sigma})$. For a given user $\mathbf{u}_j$ in the test set, $P(\mathbf{u})$ is considered to be a prior, and the objective is given by $f_j(\mathbf{m}_i) = \mathbf{m}_i^T \mathbf{u}_j$, corresponding to a GP with a linear kernel.

We cluster the movie feature vectors into $k = 80$ groups, i.e., $\mathcal{G} = \{G_1, \ldots, G_k\}$, via the $k$-means algorithm. Similarly to (26), the robust optimization problem for a given user $j$ is

$$\max_{G \in \mathcal{G}} g_j(G), \tag{31}$$

where $g_j(G) = \min_{\mathbf{m}_i \in G} f_j(\mathbf{m}_i)$. That is, for the user with feature vector $\mathbf{u}_j$, our goal is to find the group of movies to recommend such that the entire collection of movies is robust with respect to the user's preferences.

In this experiment, we compare STABLEOPT against GP-UCB and MAXIMIN-GP-UCB. We report the $\epsilon$-regret given by $g_j(G^*) - g_j(G^{(t)})$ where $G^*$ is the maximizer of (31), and $G^{(t)}$ is the reported group after time $t$. Since we are reporting groups rather than points, the baselines require slight modifications: At time $t$, GP-UCB selects the movie $\mathbf{m}_t$ (i.e., asks for the user's rating of it) and reports the group $G^{(t)}$ to which $\mathbf{m}_t$ belongs. MAXIMIN-GP-UCB reports $G^{(t)} \in \arg\max_{G \in \mathcal{G}} \min_{\mathbf{m} \in G} \mathrm{ucb}_{t-1}(\mathbf{m})$ and then selects $\mathbf{m}_t \in \arg\min_{\mathbf{m} \in G^{(t)}} \mathrm{ucb}_{t-1}(\mathbf{m})$. Finally, STABLEOPT reports a group in the same way as MAXIMIN-GP-UCB, but selects $\mathbf{m}_t$ analogously to (26). In Figure 3, we show the average $\epsilon$-regret, where the average is taken over 500 different test users. In this experiment, the average $\epsilon$-regret decreases rapidly after only a small number of rounds. Among the three methods, STABLEOPT is the only one that finds the optimal movie group.

## 6 Conclusion

We have introduced and studied a variant of GP optimization in which one requires stability/robustness to an adversarial perturbation. We demonstrated the failures of existing algorithms, and provided a new algorithm STABLEOPT that overcomes these limitations, with rigorous guarantees. We showed that our framework naturally applies to several interesting max-min optimization formulations, and we demonstrated significant improvements over some natural baselines in the experimental examples.

An interesting direction for future work is to study the $\epsilon$-stable optimization formulation in the context of hyperparameter tuning (e.g., for deep neural networks). One might expect that wide function maxima in hyperparameter space provide better generalization than narrow maxima, but establishing this requires further investigation. Similar considerations are an ongoing topic of debate in understanding the *parameter space* rather than the hyperparmeter space, e.g., see [13].

**Acknowledgment.** This work was partially supported by the Swiss National Science Foundation (SNSF) under grant number 407540_167319, by the European Research Council (ERC) under the European Union's Horizon 2020 research and innovation programme (grant agreement no725594 - time-data), by DARPA DSO's Lagrange program under grant FA86501827838, and by an NUS startup grant.

## Footnotes

[1]In this discussion, we take $d(\mathbf{x}, \mathbf{x}') = \|\mathbf{x} - \mathbf{x}'\|_2$, so that $\epsilon = 0$ recovers the standard non-stable regret [31].

[2]This is slightly different from Theorem 1, which uses the confidence bound $\mathrm{lcb}_{\tau-1}$ for $\mathbf{x}_\tau$ instead of adopting the common bound $\mathrm{lcb}_t$. We found the latter to be more suitable when the kernel hyperparameters are updated online, whereas Theorem 1 assumes a known kernel. Theorem 1 can be adapted to use $\mathrm{lcb}_t$ alone by intersecting the confidence bounds at each time instant so that they are monotonically shrinking [15].

[3]https://github.com/zi-w/Max-value-Entropy-Search

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
