[Supplementary Material]

# Supplementary Material

## Adversarially Robust Optimization with Gaussian Processes

Ilija Bogunovic, Jonathan Scarlett, Stefanie Jegelka and Volkan Cevher (NeurIPS 2018)

## A    Illustration of STABLEOPT's Execution

The following figure gives an example of the selection procedure of STABLEOPT at two different time steps:

(a) $t = 5$

(b) $t = 15$

Figure 4: An execution of STABLEOPT on the running example. We observe that after $t = 15$ steps, $\tilde{\mathbf{x}}_t$ obtained in Eq. 13 corresponds to $\mathbf{x}_\epsilon^*$.

The intermediate time steps are illustrated as follows:

(a) $t = 6$

(b) $t = 7$

(c) $t = 8$

(d) $t = 9$

(e) $t = 10$

(f) $t = 11$

(g) $t = 12$

(h) $t = 13$

(i) $t = 14$

# B  Proofs of Theoretical Results

## B.1  Proof of Theorem 1 (upper bound)

Recall that $\tilde{\mathbf{x}}_t$ is the point computed by STABLEOPT in (13) at time $t$, and that $\boldsymbol{\delta}_t$ corresponds to the perturbation obtained in STABLEOPT (Line 3) at time $t$. In the following, we condition on the event in Lemma 1 holding true, meaning that $\mathrm{ucb}_t$ and $\mathrm{lcb}_t$ provide valid confidence bounds as per (15). As stated in the lemma, this holds with probability at least $1 - \xi$.

By the definition of $\epsilon$-instant regret, we have

$$r_\epsilon(\tilde{\mathbf{x}}_t) = \max_{\mathbf{x} \in D} \min_{\boldsymbol{\delta} \in \Delta_\epsilon(\mathbf{x})} f(\mathbf{x} + \boldsymbol{\delta}) - \min_{\boldsymbol{\delta} \in \Delta_\epsilon(x_t)} f(\tilde{\mathbf{x}}_t + \boldsymbol{\delta}) \tag{32}$$

$$\leq \max_{\mathbf{x} \in D} \min_{\boldsymbol{\delta} \in \Delta_\epsilon(\mathbf{x})} f(\mathbf{x} + \boldsymbol{\delta}) - \min_{\boldsymbol{\delta} \in \Delta_\epsilon(\tilde{\mathbf{x}}_t)} \mathrm{lcb}_{t-1}(\tilde{\mathbf{x}}_t + \boldsymbol{\delta}) \tag{33}$$

$$= \max_{\mathbf{x} \in D} \min_{\boldsymbol{\delta} \in \Delta_\epsilon(\mathbf{x})} f(\mathbf{x} + \boldsymbol{\delta}) - \mathrm{lcb}_{t-1}(\tilde{\mathbf{x}}_t + \boldsymbol{\delta}_t) \tag{34}$$

$$\leq \max_{\mathbf{x} \in D} \min_{\boldsymbol{\delta} \in \Delta_\epsilon(\mathbf{x})} \mathrm{ucb}_{t-1}(\mathbf{x} + \boldsymbol{\delta}) - \mathrm{lcb}_{t-1}(\tilde{\mathbf{x}}_t + \boldsymbol{\delta}_t) \tag{35}$$

$$= \min_{\boldsymbol{\delta} \in \Delta_\epsilon(\tilde{\mathbf{x}}_t)} \mathrm{ucb}_{t-1}(\tilde{\mathbf{x}}_t + \boldsymbol{\delta}) - \mathrm{lcb}_{t-1}(\tilde{\mathbf{x}}_t + \boldsymbol{\delta}_t) \tag{36}$$

$$\leq \mathrm{ucb}_{t-1}(\tilde{\mathbf{x}}_t + \boldsymbol{\delta}_t) - \mathrm{lcb}_{t-1}(\tilde{\mathbf{x}}_t + \boldsymbol{\delta}_t) \tag{37}$$

$$= 2\beta_t^{1/2} \sigma_{t-1}(\tilde{\mathbf{x}}_t + \boldsymbol{\delta}_t), \tag{38}$$

where (33) and (35) follow from Lemma 1, (34) follows since $\boldsymbol{\delta}_t$ minimizes $\mathrm{lcb}_{t-1}$ by definition, (36) follows since $\tilde{\mathbf{x}}_t$ maximizes the robust upper confidence bound by definition, (37) follows by upper bounding the minimum by the specific choice $\boldsymbol{\delta}_t \in \Delta_\epsilon(\mathbf{x}_t)$, and (38) follows since the upper and lower confidence bounds are separated by $2\beta_t^{1/2} \sigma_{t-1}(\cdot)$ according to their definitions in (12).

In fact, the analysis from (33) to (38) shows that the following *pessimistic estimate* of $r_\epsilon(\tilde{\mathbf{x}}_t)$ is upper bounded by $2\beta_t^{1/2} \sigma_{t-1}(\tilde{\mathbf{x}}_t + \boldsymbol{\delta}_t)$:

$$\bar{r}_\epsilon(\tilde{\mathbf{x}}_t) = \max_{\mathbf{x} \in D} \min_{\boldsymbol{\delta} \in \Delta_\epsilon(\mathbf{x})} f(\mathbf{x} + \boldsymbol{\delta}) - \min_{\boldsymbol{\delta} \in \Delta_\epsilon(\tilde{\mathbf{x}}_t)} \mathrm{lcb}_{t-1}(\tilde{\mathbf{x}}_t + \boldsymbol{\delta}). \tag{39}$$

Unlike $r_\epsilon(\tilde{\mathbf{x}}_t)$, the algorithm has the required knowledge to identify the value of $t \in \{1, \dots, T\}$ with the smallest $\bar{r}_\epsilon(\tilde{\mathbf{x}}_t)$. Specifically, the first term on the right-hand side of (39) does not depend on $t$, so the smallest $\bar{r}_\epsilon(\tilde{\mathbf{x}}_t)$ is achieved by $\mathbf{x}^{(T)}$ defined in (17). Since the minimum is upper bounded by the average, it follows that

$$r_\epsilon(\mathbf{x}^{(T)}) \leq \bar{r}_\epsilon(\mathbf{x}^{(T)}) \tag{40}$$

$$\leq \frac{1}{T} \sum_{t=1}^{T} 2\beta_t^{1/2} \sigma_{t-1}(\tilde{\mathbf{x}}_t + \boldsymbol{\delta}_t) \tag{41}$$

$$\leq \frac{2\beta_T^{1/2}}{T} \sum_{t=1}^{T} \sigma_{t-1}(\tilde{\mathbf{x}}_t + \boldsymbol{\delta}_t), \tag{42}$$

where (41) uses (38), and (42) uses the monotonicity of $\beta_T$. Next, we claim that

$$2 \sum_{t=1}^{T} \sigma_{t-1}(\tilde{\mathbf{x}}_t + \boldsymbol{\delta}_t) \leq \sqrt{C_1 T \gamma_T}, \tag{43}$$

where $C_1 = 8/\log(1 + \sigma^{-2})$. In fact, this is a special case of the well-known result [31, Lemma 5.4],[4] which upper bounds the sum of posterior standard deviations of sampled points in terms of the information gain $\gamma_T$ (recall that STABLEOPT samples at location $\tilde{\mathbf{x}}_t + \boldsymbol{\delta}_t$). Combining (42)–(43) and re-arranging, we deduce that after $T$ satisfies $\frac{T}{\beta_T \gamma_T} \geq \frac{C_1}{\eta^2}$, the $\epsilon$-instant regret is at most $\eta$, thus completing the proof.

Figure 6: Illustration of functions $f_1, \dots, f_5$ equal to a common function shifted by various multiples of a given parameter $w$. In the $\epsilon$-stable setting, there is a wide region (shown in gray for the dark blue curve $f_3$) within which the perturbed function value equals $-2\eta$.

## B.2 Proof of Theorem 2 (lower bound)

Our lower bounding analysis builds heavily on that of the non-robust optimization setting with $f \in \mathcal{F}_k(B)$ studied in [27], but with important differences. Roughly speaking, the analysis of [27] is based on the difficulty of finding a very narrow "bump" of height $2\eta$ in a function whose values are mostly close to zero. In the $\epsilon$-stable setting, however, even the points around such a bump will be adversarially perturbed to another point whose function value is nearly zero. Hence, all points are essentially equally bad.

To overcome this challenge, we consider the reverse scenario: Most of the function values are still nearly zero, but there exists a narrow *valley* of depth $-2\eta$. This means that every point within an $\epsilon$-ball around the function minimizer will be perturbed to the point with value $-2\eta$. Hence, a constant fraction of the volume is still $2\eta$-suboptimal, and it is impossible to avoid this region with high probability unless the time horizon $T$ is sufficiently large. An illustration is given in Figure 6, with further details below.

We now proceed with the formal proof.

### B.2.1 Preliminaries

Recall that we are considering an arbitrary given (deterministic) GP optimization algorithm. More precisely, such an algorithm consists of a sequence of decision functions that return a sampling location $\mathbf{x}_t$ based on $y_1, \dots, y_{t-1}$, and an additional decision function that reports the final point $\mathbf{x}^{(T)}$ based on $y_1, \dots, y_T$. The points $\mathbf{x}_1, \dots, \mathbf{x}_{t-1}$ (or $\mathbf{x}_1, \dots, \mathbf{x}_T$) do not need to be treated as additional inputs to these functions, since $(\mathbf{x}_1, \dots, \mathbf{x}_{t-1})$ is a deterministic function of $(y_1, \dots, y_{t-1})$.

We first review several useful results and techniques from [27]:

- We lower bound the worst-case $\epsilon$-regret within $\mathcal{F}_k(B)$ by the $\epsilon$-regret averaged over a suitably-designed finite collection $\{f_1, \dots, f_M\} \subset \mathcal{F}_k(B)$ of size $M$.

- We choose each $f_m(\mathbf{x})$ to be a shifted version of a common function $g(\mathbf{x})$ on $\mathbb{R}^p$. Specifically, each $f_m(\mathbf{x})$ is obtained by shifting $g(\mathbf{x})$ by a different amount, and then cropping to $D = [0, 1]^p$. For our purposes, we require $g(\mathbf{x})$ to satisfy the following properties:

  1. The RKHS norm in $\mathbb{R}^p$ is bounded, $\|g\|_k \leq B$;

  2. We have (i) $g(\mathbf{x}) \in [-2\eta, 2\eta]$ with minimum value $g(0) = -2\eta$, and (ii) there is a "width" $w$ such that $g(\mathbf{x}) > -\eta$ for all $\|\mathbf{x}\|_\infty \geq w$;

  3. There are absolute constants $h_0 > 0$ and $\zeta > 0$ such that $g(\mathbf{x}) = \frac{2\eta}{h_0} h\left(\frac{\mathbf{x}\zeta}{w}\right)$ for some function $h(\mathbf{z})$ that decays faster than any finite power of $\|\mathbf{z}\|_2^{-1}$ as $\|\mathbf{z}\|_2 \to \infty$.

Letting $g(\mathbf{x})$ be such a function, we construct the $M$ functions by shifting $g(\mathbf{x})$ so that each $f_m(\mathbf{x})$ is centered on a unique point in a uniform grid, with points separated by $w$ in each dimension. Since $D = [0,1]^p$, one can construct

$$M = \left\lfloor \left(\frac{1}{w}\right)^p \right\rfloor \tag{44}$$

such functions. We will use this construction with $w \ll 1$, so that there is no risk of having $M = 0$, and in fact $M$ can be assumed larger than any desired absolute constant.

- It is shown in [27] that the above properties[5] can be achieved with

$$M = \left\lfloor \left( \frac{r\sqrt{\log \frac{B(2\pi l^2)^{p/4} h(0)}{2\eta}}}{\zeta \pi l} \right)^p \right\rfloor \tag{45}$$

in the case of the SE kernel, and with

$$M = \left\lfloor \left(\frac{Bc_3}{\eta}\right)^{p/\nu} \right\rfloor \tag{46}$$

in the case of the Matérn kernel, where

$$c_3 := \left(\frac{r}{\zeta}\right)^\nu \cdot \left( \frac{c_2^{-1/2}}{2(8\pi^2)^{(\nu+p/2)/2}} \right), \tag{47}$$

and where $c_2 > 0$ is an absolute constant. Note that these values of $M$ amount to choosing $w$ in (44), and the assumption of sufficiently small $\frac{\eta}{B}$ in the theorem statement ensures that $M \gg 1$ (or equivalently $w \ll 1$) as stated above.

- Property 2 above ensures that the "robust" function value $\min_{\boldsymbol{\delta} \in \Delta_\epsilon(\mathbf{x})} f(\mathbf{x})$ equals $-2\eta$ for any $\mathbf{x}$ whose $\epsilon$-neighborhood includes the minimizer $\mathbf{x}_{\min}$ of $f$, while being $-\eta$ or higher for any input whose entire $\epsilon$-neighborhood is separated from $\mathbf{x}_{\min}$ by at least $w$. For $w \ll 1$ and $\epsilon < 0.5$, a point of the latter type is guaranteed to exist, which implies

$$r_\epsilon(\mathbf{x}) \geq \eta \tag{48}$$

for any $\mathbf{x}$ whose $\epsilon$-neighborhood includes $\mathbf{x}_{\min}$.

In addition, we introduce the following notation, also used in [27]:

- The probability density function of the output sequence $\mathbf{y} = (y_1, \ldots, y_T)$ when the underlying function is $f_m$ is denoted by $P_m(\mathbf{y})$. We also define $f_0(\mathbf{x}) = 0$ to be the zero function, and define $P_0(\mathbf{y})$ analogously for the case that the optimization algorithm is run on $f_0$. Expectations and probabilities (with respect to the noisy observations) are similarly written as $\mathbb{E}_m, \mathbb{P}_m, \mathbb{E}_0$, and $\mathbb{P}_0$ when the underlying function is $f_m$ or $f_0$. On the other hand, in the absence of a subscript, $\mathbb{E}[\cdot]$ and $\mathbb{P}[\cdot]$ are taken with respect to the noisy observations *and* the random function $f$ drawn uniformly from $\{f_1, \ldots, f_M\}$ (recall that we are lower bounding the worst case by this average).

- Let $\{\mathcal{R}_m\}_{m=1}^M$ be a partition of the domain into $M$ regions according the above-mentioned uniform grid, with $f_m$ taking its minimum value of $-2\eta$ in the centre of $\mathcal{R}_m$. Moreover, let $j_t$ be the index at time $t$ such that $\mathbf{x}_t$ falls into $\mathcal{R}_{j_t}$; this can be thought of as a quantization of $\mathbf{x}_t$.

- Define the maximum (absolute) function value within a given region $\mathcal{R}_j$ as

$$\overline{v}_m^j := \max_{\mathbf{x} \in \mathcal{R}_j} |f_m(\mathbf{x})|, \tag{49}$$

and the maximum KL divergence to $P_0$ within the region as

$$\overline{D}_m^j := \max_{\mathbf{x} \in \mathcal{R}_j} D(P_0(\cdot|\mathbf{x}) \| P_m(\cdot|\mathbf{x})), \tag{50}$$

where $P_m(y|\mathbf{x})$ is the distribution of an observation $y$ for a given selected point $\mathbf{x}$ under the function $f_m$, and similarly for $P_0(y|\mathbf{x})$.

- Let $N_j \in \{0, \ldots, T\}$ be a random variable representing the number of points from $\mathcal{R}_j$ that are selected throughout the $T$ rounds.

Next, we present several useful lemmas. The following well-known change-of-measure result, which can be viewed as a form of Le Cam's method, has been used extensively in both discrete and continuous bandit problems.

**Lemma 2.** [1, p. 27] *For any function $a(\mathbf{y})$ taking values in a bounded range $[0, A]$, we have*

$$\left| \mathbb{E}_m[a(\mathbf{y})] - \mathbb{E}_0[a(\mathbf{y})] \right| \leq A \, d_{\mathrm{TV}}(P_0, P_m) \tag{51}$$

$$\leq A \sqrt{D(P_0 \| P_m)}, \tag{52}$$

*where $d_{\mathrm{TV}}(P_0, P_m) = \frac{1}{2} \int_{\mathbb{R}^T} |P_0(\mathbf{y}) - P_m(\mathbf{y})| \, d\mathbf{y}$ is the total variation distance.*

We briefly remark on some slight differences here compared to [1, p. 27]. There, only $\mathbb{E}_m[a(\mathbf{y})] - \mathbb{E}_0[a(\mathbf{y})]$ is upper bounded in terms of $d_{\mathrm{TV}}(P_0, P_m)$, but one easily obtains the same upper bound on $\mathbb{E}_0[a(\mathbf{y})] - \mathbb{E}_m[a(\mathbf{y})]$ by interchanging the roles of $P_0$ and $P_m$. The step (52) follows from Pinsker's inequality, $d_{\mathrm{TV}}(P_0, P_m) \leq \sqrt{\frac{D(P_0 \| P_m)}{2}}$, and by upper bounding $\frac{1}{\sqrt{2}} \leq 1$ to ease the notation.

The following result simplifies the divergence term in (52).

**Lemma 3.** [27, Eq. (44)] *Under the preceding definitions, we have*

$$D(P_0 \| P_m) \leq \sum_{j=1}^{M} \mathbb{E}_0[N_j] \overline{D}_m^j. \tag{53}$$

The following well-known property gives a formula for the KL divergence between two Gaussians.

**Lemma 4.** [27, Eq. (36)] *For $P_1$ and $P_2$ being Gaussian with means $(\mu_1, \mu_2)$ and a common variance $\sigma^2$, we have*

$$D(P_1 \| P_2) = \frac{(\mu_1 - \mu_2)^2}{2\sigma^2}. \tag{54}$$

Finally, we have the following technical result regarding the "needle-in-haystack" type function constructed above.

**Lemma 5.** [27, Lemma 7] *The functions $\{f_m\}_{m=1}^M$ corresponding to (45)–(46) are such that the quantities $\overline{v}_m^j$ satisfy $\sum_{m=1}^M (\overline{v}_m^j)^2 = O(\eta^2)$ for all $j$.*

### B.2.2 Analysis of the average $\epsilon$-stable regret

Let $J_{\mathrm{bad}}(m)$ be the set of $j$ such that all $\mathbf{x} \in \mathcal{R}_j$ yield $\min_{\boldsymbol{\delta} \in \Delta_\epsilon(\mathbf{x})} f(\mathbf{x} + \boldsymbol{\delta}) = -2\eta$ when the true function is $f_m$, and define $\mathcal{R}_{\mathrm{bad}}(m) = \cup_{j \in J_{\mathrm{bad}}(m)} \mathcal{R}_j$. By the $\epsilon$-regret lower bound in (48), we have

$$\mathbb{E}_m[r_\epsilon(\mathbf{x}^{(T)})] \geq \eta \mathbb{P}_m[\mathbf{x}^{(T)} \in \mathcal{R}_{\mathrm{bad}}(m)] \tag{55}$$

$$\geq \eta \left( \mathbb{P}_0[\mathbf{x}^{(T)} \in \mathcal{R}_{\mathrm{bad}}(m)] - \sqrt{D(P_0 \| P_m)} \right) \tag{56}$$

$$\geq \eta \left( \mathbb{P}_0[\mathbf{x}^{(T)} \in \mathcal{R}_{\mathrm{bad}}(m)] - \sqrt{\sum_{j=1}^M \mathbb{E}_0[N_j] \overline{D}_m^j} \right), \tag{57}$$

where (56) follows from Lemma 2 with $a(\mathbf{y}) = \mathbf{1}\{\mathbf{x}^{(T)} \in \mathcal{R}_{\mathrm{bad}}(m)\}$ and $A = 1$ (recall that $\mathbf{x}^{(T)}$ is a function of $\mathbf{y} = (y_1, \ldots, y_T)$), and (57) follows from Lemma 3. Averaging over $m$ uniform on $\{1, \ldots, M\}$, we obtain

$$\mathbb{E}[r_\epsilon(\mathbf{x}^{(T)})] \geq \eta \left( \frac{1}{M} \sum_{m=1}^M \mathbb{P}_0[\mathbf{x}^{(T)} \in \mathcal{R}_{\mathrm{bad}}(m)] - \frac{1}{M} \sum_{m=1}^M \sqrt{\sum_{j=1}^M \mathbb{E}_0[N_j] \overline{D}_m^j} \right). \tag{58}$$

We proceed by bounding the two terms separately.

- We first claim that

$$\frac{1}{M} \sum_{m=1}^{M} \mathbb{P}_0[\mathbf{x}^{(T)} \in \mathcal{R}_{\text{bad}}(m)] \geq C_1 \qquad (59)$$

for some $C_1 > 0$. To show this, it suffices to prove that any given $\mathbf{x}^{(T)} \in D$ is in at least a constant fraction of the $\mathcal{R}_{\text{bad}}(m)$ regions, of which there are $M$. This follows from the fact that the $\epsilon$-ball centered at $\mathbf{x}_{m,\min} = \arg\min_{\mathbf{x} \in D} f_m(\mathbf{x})$ takes up a constant fraction of the volume of $D$, where the constant depends on both the stability parameter $\epsilon$ and the dimension $p$. A small caveat is that because the definition of $\mathcal{R}_{\text{bad}}$ insists that the *every* point in the region $\mathcal{R}_j$ is within distance $\epsilon$ of $\mathbf{x}_{m,\min}$, the left-hand side of (59) may be slightly below the relevant ratio of volumes above. However, since Theorem 2 assumes that $\frac{\eta}{B}$ is sufficiently small, the choices of $M$ in (45) and (46) ensure that $M$ is sufficiently large for this "quantization" effect to be negligible.

- For the second term in (58), we claim that

$$\frac{1}{M} \sum_{m=1}^{M} \sqrt{\sum_{j=1}^{M} \mathbb{E}_0[N_j]\overline{D}_m^j} \leq C_2 \frac{\eta}{\sigma}\sqrt{\frac{T}{M}} \qquad (60)$$

for some $C_2 > 0$. To see this, we write

$$\frac{1}{M} \sum_{m=1}^{M} \sqrt{\sum_{j=1}^{M} \mathbb{E}_0[N_j]\overline{D}_m^j}$$

$$= O\left(\frac{1}{\sigma}\right) \cdot \frac{1}{M} \sum_{m=1}^{M} \sqrt{\sum_{j=1}^{M} \mathbb{E}_0[N_j](\overline{v}_m^j)^2} \qquad (61)$$

$$\leq O\left(\frac{1}{\sigma}\right) \cdot \sqrt{\frac{1}{M} \sum_{m=1}^{M} \sum_{j=1}^{M} \mathbb{E}_0[N_j](\overline{v}_m^j)^2} \qquad (62)$$

$$= O\left(\frac{1}{\sigma}\right) \cdot \sqrt{\frac{1}{M} \sum_{j=1}^{M} \mathbb{E}_0[N_j]\left(\sum_{m=1}^{M} (\overline{v}_m^j)^2\right)} \qquad (63)$$

$$= O\left(\frac{\eta}{\sqrt{M}\sigma}\right) \cdot \sqrt{\sum_{j=1}^{M} \mathbb{E}_0[N_j]} \qquad (64)$$

$$= O\left(\frac{\sqrt{T}\eta}{\sqrt{M}\sigma}\right), \qquad (65)$$

where (61) follows since the divergence $D(P_0(\cdot|\mathbf{x})\|P_m(\cdot|\mathbf{x}))$ associated with a point $\mathbf{x}$ having value $v(\mathbf{x})$ is $\frac{v(\mathbf{x})^2}{2\sigma^2}$ (cf., (54)), (62) follows from Jensen's inequality, (64) follows from Lemma 5, and (65) follows from $\sum_j N_j = T$.

Substituting (59) and (60) into (58), we obtain

$$\mathbb{E}[r_\epsilon(\mathbf{x}^{(T)})] \geq \eta\left(C_1 - C_2\frac{\eta}{\sigma}\sqrt{\frac{T}{M}}\right), \qquad (66)$$

which implies that the regret is lower bounded by $\Omega(\eta)$ unless $T = \Omega\left(\frac{M\sigma^2}{\eta^2}\right)$. Substituting $M$ from (45) and (46), we deduce that the conditions on $T$ in the theorem statement are necessary to achieve average regret $\mathbb{E}[r_\epsilon(\mathbf{x}^{(T)})] = O(\eta)$ with a sufficiently small implied constant.

### B.2.3 From average to high-probability regret

Recall that we are considering functions whose values lie in the range $[-2\eta, 2\eta]$, implying that $r_\epsilon(\mathbf{x}^{(T)}) \leq 4\eta$. Letting $T_\eta$ be the lower bound on $T$ derived above for achieving average regret

$O(\eta)$ (i.e., we have $\mathbb{E}[r_\epsilon^{(T_\eta)}] = \Omega(\eta)$), it follows from the reverse Markov inequality (i.e., Markov's inequality applied to the random variable $4\eta - r_\epsilon^{(T_\eta)}$) that

$$\mathbb{P}[r_\epsilon(\mathbf{x}^{(T_\eta)}) \geq c\eta] \geq \frac{\Omega(\eta) - c\eta}{4\eta - c\eta} \tag{67}$$

for any $c > 0$ sufficiently small for the numerator and denominator to be positive. The right-hand side is lower bounded by a constant for any such $c$, implying that the probability of achieving $\epsilon$-regret at most $c\eta$ cannot be arbitrarily close to one. By renaming $c\eta$ as $\eta'$, it follows that in order to achieve some target $\epsilon$-stable regret $\eta'$ with probability sufficiently close to one, a lower bound of the same form as the average regret bound holds. In other words, the conditions on $T$ in the theorem statement remain necessary also for the high-probability regret.

We emphasize that Theorem 2 concerns the high-probability regret when "high probability" means *sufficiently close to one* as a function of $\epsilon$, $p$, and the kernel parameters (but still constant with respect to $T$ and $\eta$). We do not claim a lower bound under any particular *given* success probability (e.g., $\eta$-optimality with probability at least $\frac{3}{4}$).

# C   Details on Variations from Section 4

We claim that the STABLEOPT variations and theoretical results outlined in Section 4 are in fact special cases of Algorithm 1 and Theorem 1, despite being seemingly quite different. The idea behind this claim is that Algorithm 1 and Theorem 1 allow for the "distance" function $d(\cdot, \cdot)$ to be completely arbitrary, so we may choose it in rather creative/unconventional ways.

In more detail, we have the following:

- For the unknown parameter setting $\max_{\mathbf{x} \in D} \min_{\boldsymbol{\theta} \in \Theta} f(\mathbf{x}, \boldsymbol{\theta})$, we replace $\mathbf{x}$ in the original setting by the concatenated input $(\mathbf{x}, \boldsymbol{\theta})$, and set

$$d((\mathbf{x}, \boldsymbol{\theta}), (\mathbf{x}', \boldsymbol{\theta}')) = \|\mathbf{x} - \mathbf{x}'\|_2. \tag{68}$$

  If we then set $\epsilon = 0$, we find that the input $\mathbf{x}$ experiences no perturbation, whereas $\boldsymbol{\theta}$ may be perturbed arbitrarily, thereby reducing (7) to $\max_{\mathbf{x} \in D} \min_{\boldsymbol{\theta} \in \Theta} f(\mathbf{x}, \boldsymbol{\theta})$ as desired.

- For the robust estimation setting, we again use the concatenated input $(\mathbf{x}, \boldsymbol{\theta})$. To avoid overloading notation, we let $d_0(\boldsymbol{\theta}, \boldsymbol{\theta}')$ denote the distance function (applied to $\boldsymbol{\theta}$ alone) adopted for this case in Section 4. We set

$$d((\mathbf{x}, \boldsymbol{\theta}), (\mathbf{x}', \boldsymbol{\theta}')) = \begin{cases} d_0(\boldsymbol{\theta}, \boldsymbol{\theta}') & \mathbf{x} = \mathbf{x}' \\ \infty & \mathbf{x} \neq \mathbf{x}'. \end{cases} \tag{69}$$

  Due to the second case, the input $\mathbf{x}$ experiences no perturbation, since doing so would violate the distance constraint of $\epsilon$. We are then left with $\mathbf{x} = \mathbf{x}'$ and $d_0(\boldsymbol{\theta}, \boldsymbol{\theta}') \leq \epsilon$, as required.

- For the grouped setting $\max_{G \in \mathcal{G}} \min_{\mathbf{x} \in G} f(\mathbf{x})$, we adopt the function

$$d(\mathbf{x}, \mathbf{x}') = \mathbf{1}\{\mathbf{x} \text{ and } \mathbf{x}' \text{ are in different groups}\}, \tag{70}$$

  and set $\epsilon = 0$. Considering the formulation in (7), we find that any two inputs $\mathbf{x}$ and $\mathbf{x}'$ yield the same $\epsilon$-stable objective function, and hence, reporting a point $\mathbf{x}$ is equivalent to reporting its group $G$. As a result, (7) reduces to the desired formulation $\max_{G \in \mathcal{G}} \min_{\mathbf{x} \in G} f(\mathbf{x})$.

The variations of STABLEOPT described in (20)–(26), as well as the corresponding theoretical results outlined in Section 4, follow immediately by substituting the respective choices of $d(\cdot, \cdot)$ and $\epsilon$ above into Algorithm 1 and Theorem 1. It should be noted that in the first two examples, the definition of $\gamma_t$ in (14) is modified to take the maximum over not only $\mathbf{x}_1, \cdots, \mathbf{x}_t$, but also $\boldsymbol{\theta}_1, \cdots, \boldsymbol{\theta}_t$.

# D   Lake Data Experiment

We consider an application regarding environmental monitoring of inland waters, using a data set containing 2024 in situ measurements of chlorophyll concentration within a vertical transect plane, collected by an autonomous surface vessel in Lake Zürich. This data set was considered in previous

(a) Chlorophyll concentration       (b) Robust objective       (c) $\epsilon$-regret

Figure 7: Experiment on the Zürich lake dataset; In the later rounds STABLEOPT is the only method that reports a near-optimal $\epsilon$-stable point.

works such as [7, 15] to detect regions of high concentration. In these works, the goal was to locate all regions whose concentration exceeds a pre-defined threshold.

Here we consider a different goal: We seek to locate a region of a given size such that the concentration throughout the region is as high as possible (in the max-min sense). This is of interest in cases where high concentration only becomes relevant when it is spread across a sufficiently wide area. We consider rectangular regions with different pre-specified lengths in each dimension:

$$\Delta_{\epsilon_D, \epsilon_L}(\mathbf{x}) = \{\mathbf{x}' - \mathbf{x} : \mathbf{x}' \in D, \ |x_D - x'_D| \leq \epsilon_D \cap |x_L - x'_L| \leq \epsilon_L\}, \tag{71}$$

where $\mathbf{x} = (x_D, x_L)$ and $\mathbf{x}' = (x'_D, x'_L)$ indicate the depth and length, and we denote the corresponding stability parameters by $(\epsilon_D, \epsilon_L)$. This corresponds to $d(\cdot, \cdot)$ being a weighted $\ell_\infty$-norm.

We evaluate each algorithm on a $50 \times 50$ grid of points, with the corresponding values coming from the GP posterior that was derived using the original data. We use the Matérn-5/2 ARD kernel, setting its hyperparameters by maximizing the likelihood on a second (smaller) available dataset. The parameters $\epsilon_D$ and $\epsilon_L$ are set to 1.0 and 100.0, respectively. The stability requirement changes the global maximum and its location, as can be observed in Figure 7. The number of sampling rounds is $T = 120$, and each algorithm is initialized with the same 10 random data points and corresponding observations. The performance is averaged over 100 different runs, where every run corresponds to a different random initialization. In this experiment, STABLE-GP-UCB achieves the smallest $\epsilon$-regret in the early rounds, while in the later rounds STABLEOPT is the only method that reports a near-optimal $\epsilon$-stable point.

## Footnotes

[4]More precisely, [31, Lemma 5.4] alongside an application of the Cauchy-Schwarz inequality as in [31].

[5]Here $g(\mathbf{x})$ plays the role of $-g(\mathbf{x})$ in [27] due to the discussion at the start of this appendix, but otherwise the construction is identical.