[Reviews · NeurIPS 2018]

Reviewer 1



-- Paper Summary This paper considers an optimisation set-up where the target function may not be known exactly, but rather may be prone to adversarial attacks and/or uncertainty. This interpretation can also be adapted to other problem settings such as reporting optimal subsets/groups of points rather than individual points. The bounds considered in this work are backed up by both theoretic proofs and a thorough set of varied experiments. -- Originality + Significance The work presented in this paper appears to be sufficiently novel within the context of GP optimization. The proposal is very well-motivated in the introduction, while the varied experiments are effective in showcasing the potential use-cases of this method. This paper is indeed not the first to consider robust optimisation, but it is sufficiently differentiated (adversarial rather than random uncertainty, noisy function evaluations, single-point sampling rather than batch). Related work is also appropriately cited and discussed. Given its inherent similarity to optimising under uncertainty (albeit not adversarial), the connection to Bayesian optimisation could perhaps be expanded further; it is currently only briefly mentioned in passing. With regards to significance, I think that the appeal of this paper is not limited to the optimisation community but could also be of interest to other machine learning researchers and practitioners. -- Quality I am satisfied with the extent of both the theoretic analysis and experimental evaluation. While I look forward to reading the opinion of other reviewers who work more closely in the field of multi-armed bandit optimization, I currently have very little concerns about this paper. The performance of the algorithm with respect to different configurations of \beta and \epsilon could be explored further using synthetic examples (even if only in the supplementary material). Although the authors claim that their assignment is typically dependent on the problem being targeted, setting these parameters still feels like an overly ambiguous task. The theoretical choice is claimed to be generally unsuitable and is consequently replaced using a fairly vague heuristic given in L220-222. It might be down to wording, but it is also slightly unclear as to whether the baselines considered are indeed the closest competitors to the method developed in the paper or simply base approaches which the proposal is all but guaranteed to outperform. Several papers are mentioned in the related work section but are not directly considered in the experimental evaluation. In this regard, I would encourage the authors to justify their choice of competitors more effectively. -- Writing and Clarity This is undeniably a very well-written paper – I could not spot any typos while reading it, and all the concepts are presented in a clear and concise manner. The balance between the content that is included in the main paper and that which is left to the supplementary material is also very suitable – the supplementary material contains additional experiments as well as proofs corroborating the theorems presented in the main paper. -- Overall Recommendation While similar ideas to those explored in this paper may have been tangentially investigated in other forms, I believe this is an excellent paper which ticks all the boxes with regards to content, evaluation and overall presentation. I believe that such a paper could be very well-received by the optimisation community participating at NIPS. -------------- -- Post rebuttal Reading the other reviews and the rebuttal has consolidated my views on the quality and significance of this paper. I highly encourage the authors to incorporate the suggestions pointed out in the reviews and subsequently addressed in the rebuttal, in particular the connection to Bayesian optimisation and clarifying why ES and EI share the same issues as GP-UCB. Great work!

Reviewer 2



The authors consider a problem of black-box optimisation with additional robustness requirements. As opposed to standard Bayesian optimisation, where the unknown function f is modelled typically with GP and new evaluation location chosen to improve the knowledge of the global optimiser sequentially, the authors instead want to find a point whose objective function value remains almost optimal even after some perturbation of the point. In this setting e.g. a global optimiser that lies on top of a narrow peak might be suboptimal choice because even a small change of the point could change its value drastically while a point having smaller objective function value but lying in a more flat region might be better choice. The authors formulate the problem so that instead of trying to find a global optimiser, they look for a point that is the maximiser of the minimum of f over some \epsilon-neighbourhood of the point. They extend the well-known GP-UCB/LCB strategy to this robust setting and provide both theoretical and empirical support for their method. Also, they show that many "max-min" type of optimisation problems in literature can be seen as special cases of their approach. This is a very strong and well-written paper on an interesting problem and, as far as I know, the topic has not been studied in such detail previously. The authors also list some practical applications for their method. While I consider myself as an expert in BayesOpt and related methods and could understand the paper quite well, I am not familiar with the underlying theoretical framework (based on e.g. [30]) and could not check the proofs of the theoretical results. However, the theoretical results also look carefully done. I have the following minor concerns and comments/questions of the paper: - I found no discussion on how the optimisation of (10) is carried out and what is the computational cost as compared to standard BayesOpt methods like GP-UCB or the time required to evaluate f. Optimisation of (10) may not be straightforward because the nested minimisation over the \epsilon ball. Perhaps these computational aspects could be briefly discussed in the supplement. Experiments focus on rather low-dimensional cases where grid optimisation might already work just fine but I also wonder how would the approach work in high-dimensions? - There appears to be large difference between GP-UCB and STABLE-GP-UCB in Fig2c and especially in Fig7c although the only difference between these two appears to be the selection of the reported point. I thus wonder if MAXIMIN-GP-UCB would work better if the reported point was chosen similarly as with STABLE-* methods. - lines 118-220: Generally, I am a bit puzzled on how the reported point should be chosen in the first place. Maybe I am missing something here but why not just use the GP mean function as a proxy to the unknown function f and then use (5) to select the final reported point? Or is it author's experience that the (pessimistic) estimate based on the lcb criteria actually works better in practice? - line 225: Comparison to only GP-UCB-based methods seems reasonable but I believe that one could also extend EI and entropy search to this robust setting (evaluating the resulting acquisition function might require numerical computations though). - line 221: Choosing \beta_t^{1/2}=2 seems a bit arbitrary choice and I wonder what is actually going on here. I could imagine that a more conservative strategy that produces more exploration of the parameter space might actually be useful here. - Distance d(x,x') is also an input of Algorithm 1, right? - I wonder what GP model was used in Robust robot pushing experiment with the two functions f_i, i=1,2? - For consistency, the order of the methods in the legend box of Fig3 could be the same as in other Figs. Edit after author rebuttal: After reading the rebuttal and other reviews I remain confident that this is high quality work and should be accepted. However, I am still a bit puzzled about some design choices and how they would affect the experimental results in practice, so I hope the authors can indeed clarify these in the final version.

Reviewer 3



[edit] I found the rebuttal to my points fairly brief and unsatisfying - if the performance of an overall system is improved by using a less principled component, or a component without various proofs associated, that's an interesting result. That said, I still think this paper is interesting and worthy of inclusion. [/edit] The paper presents a method of improving the stability of Bayesian optimisation by selecting a sample by: 1. setting x maximise min_d(ucb(x+d)) for some distance d in a constrained region around the intended sample location 2. setting x_sample = x + d_s by min_{d_s}lcb(x + d_s) This attempts to make the modelling robust to the worst case maximally adversarial perturbation. The two step procedure first finds a robust optimiser, then actually samples the most pessimistic point in the neighbourhood of this point. The reported point is the the point in the samples with the __highest__ pessimistic lcb - the end result being that the reported point is expected to be most stable. The authors relate their work qualitatively other approaches to robust BO in the literature - particularly "Bayesian Optimization Under Uncertainty" by Beland and Nair, however do not perform experiments comparing their results to those of the other papers. As similar in spirit approach not mentioned is presented in "Unscented Bayesian Optimization for Safe Robot Grasping" by Nogueira et al - there, random noise is added to the inputs of the sample, and propagated into the prediction using the unscented transform. In the paper under review and the two mentioned, the end result is that the optimiser avoids sharp optima. I was unclear as to the meaning of the line 42 comment "and applied to different points" - I would appreciate further explanation in the author's rebuttal. My understanding of that paper is not deep so I may have missed some key difference, but it seems like the formulation used to optimise the Branin function would be applicable to this task (if suboptimal). "Overall score" My overall score reflects an average over the high quality of the presentation and exposition in the first part (an 8) and the quality of the experiments (a 5 or 6). The authors reject comparisons to "Bayesian Optimization Under Uncertainty", however it appears that the numerical studies in section 3 of that paper on the Branin function cover a very similar problem, albeit without the __adversarial__ perturbation. I would also have appreciated a comparison to "Unscented Bayesian Optimization for Safe Robot Grasping". These experiments would have justified the additional machinery introduced in this paper. Additionally, although less importantly, I would have appreciated a comparison to other robust techniques not based around BO (e.g. line 52's '[5, 9, 19, 31, 36]') - again, do we need the additional machinery introduced by BO to handle these cases? "Confidence score" I am fairly knowledgeable on the subject of BO, however my knowledge of robust optimisation is lacking.